# ABO Incompatible Liver Transplantation in Children: A 20 Year Experience from Centres in the TransplantChild European Reference Network

**DOI:** 10.3390/children8090760

**Published:** 2021-08-31

**Authors:** Małgorzata Markiewicz-Kijewska, Piotr Kaliciński, Juan Torres Canizales, Angelo Di Giorgio, Ulrich Baumann, Carl Jorns, Alastair Baker, Maria Francelina Lopes, Esteban Frauca Remacha, Eduardo Lopez-Granados, Paloma Jara Vega, Maria-Sole Basso, Grzegorz Kowalewski, Diana Kamińska, Sandra Ferreira, Daniela Liccardo, Andrea Pietrobattista, Marco Spada

**Affiliations:** 1Department of Pediatric Surgery and Organ Transplantation, Children’s Memorial Health Institute, 04-730 Warsaw, Poland; m.markiewicz@ipczd.pl (M.M.-K.); g.kowalewski@ipczd.pl (G.K.); 2Center for Biomedical Network Research on Rare Diseases (CIBERER U767), Lymphocyte Pathophysiology in Immunodeficiencies Group, Immunology Unit, La Paz Institute of Biomedical Research (IdiPAZ), La Paz University Hospital, 28046 Madrid, Spain; jmtorres@clinic.cat (J.T.C.); elgranados@salud.madrid.org (E.L.-G.); 3Department of Pediatric Hepatology, Gastroenterology and Transplantation, ASST Hospital Papa Giovanni XXIII, 24127 Bergamo, Italy; adigiorgio@asst-pg23.it; 4Division of Pediatric Gastroenterology and Hepatology, Hannover Medical School, 30625 Hannover, Germany; Baumann.U@mh-hannover.de; 5Department of Transplantation Surgery, Karolinska University Hospital, 171 76 Stockholm, Sweden; carl.jorns@sll.se; 6Pediatric Liver, Gastrointestinal and Nutrition Centre, King’s College London School of Medicine, King’s College Hospital, Denmark Hill, London SE5 9RS, UK; alastair.baker@nhs.net; 7Department of Pediatric Surgery, Centro de Investigação e Formação Clínica, Hospital Pediátrico, Centro Hospitalar e Universitário de Coimbra, Faculty of Medicine, University of Coimbra, 3000-075 Coimbra, Portugal; mfrancelina@yahoo.com; 8Servicio de Hepatología Pediátrica, Hospital Universitario La Paz, 28046 Madrid, Spain; esteban.frauca@salud.madrid.org (E.F.R.); paloma.jara@transplantchild.eu (P.J.V.); 9Department of Hepatology, Gastroenterology and Nutrition, Ospedale Pediatrico Bambino Gesu, 00165 Roma, Italy; msole.basso@opbg.net (M.-S.B.); daniela.liccardo@opbg.net (D.L.); andrea.pietrobattista@opbg.net (A.P.); 10The Department of Gastroenterology, Hepatology, Nutrition Disorder and Pediatric, The Children’s Memorial Health Institute, 04-730 Warsaw, Poland; d.kaminska@ipczd.pl; 11Hepatology and Pediatric Liver Transplantation Unit, Centro Hospitalar e Universitário de Coimbra, 3000-075 Coimbra, Portugal; sandraferreira@chuc.min-saude.pt; 12Department of Abdominal Transplantation and Hepatobiliopancreatic Surgery, Bambino Gesù Children’s Hospital IRCCS, 00165 Rome, Italy; marco.spada@opbg.net

**Keywords:** AB0-incompatible liver transplantation, children, immunosuppression, rejection, complications, patient survival, graft survival

## Abstract

An increasing number of AB0-incompatible (AB0i) liver transplantations (LT) are being undertaken internationally in recent years due to organ shortages and the need for urgent transplantation. The aim of our study was establish the value of ABOi LT from available retrospective results of AB0i pediatric liver transplantations performed in European reference centers now belonging to the TransplantChild, European Reference Network (ERN). Data from medical records were analyzed, including demographic data, diagnosis, urgency of transplantation, time on the waiting list, PELD/MELD score, desensitization procedures, immunosuppression, selected post-transplant complications, and patient and graft survival. A total of 142 patients (pts) with transplants between 1986 and 2018 in 8 European transplant centers were included in the study. The indications for liver transplantation were: cholestatic diseases in 62 pts, acute liver failure in 42 pts, and other conditions in the remaining 38 pts. Sixty-six patients received grafts from living donors, and seventy-six received grafts from deceased donors. Both patient and graft survival were significantly affected by deceased donor type, urgent transplantation, and the development of vascular complications. In the multivariate analysis, vascular complications had a negative impact on patient and graft survival, while a longer time from the first AB0i LT in the study showed better results, suggesting an international learning experience. In conclusion, we believe that AB0i LT in children is now a safe procedure that may be adopted more readily in children.

## 1. Introduction

For more than 30 years, liver transplantation (LT) has been accepted as a major life-saving yet routine treatment for children with end-stage liver disease. This highly successful situation has led to an international situation of organ shortages that is especially critical for patients needing urgent transplantation. Therefore, a number of responses to the shortage have been adopted including marginal donors and donors with recipient-incompatible blood groups (AB0i).

Starzl et al. were the first to report 11 cases of AB0i LT in 1979 [1]. Initially, AB0i LT was considered only for urgent transplantations in patients with acute liver failure or for re-transplantation in cases of early graft loss due to hepatic artery thrombosis (HAT) or graft primary non-function. With developments in immunosuppression including new induction protocols to reduce anti-AB isoagglutinin titers, the results of AB0i organ transplantations improved. Various desensitization protocols including splenectomy, high doses of immunoglobulins (1–2 g/kg body mass), plasmapheresis, immunoadsorption, rituximab, and basiliximab as single or combined treatment were introduced to protect AB0i grafts against humoral rejection: with varying success rates reported [1,2,3,4].

It has been noted by several authors that the results of AB0i LT were better in the pediatric recipient population, particularly among the youngest children below 2 years of age, attributed to their immature immune system [5,6].

The aim of our study was to retrospectively review the experience of AB0i transplants in children in leading pediatric liver transplant centers of the TransplantChild, European Reference Network in Europe. We set out to review the indications, practices for pre- and post-transplant treatment, and the results of these transplantations with the inclusion of immunological and other complications to assess patients and graft survival as well as the factors influencing these results. We also wanted to assess if any progress has been made in the outcomes of AB0i LT in children over the study period and to consider the current situation with respect to the use of this organ source.

## 2. Materials and Methods

Clinical material: data on 150 pts were collected from eight European transplant centers. Eight patients were excluded from the analysis due to a lack of sufficient data. The contributing centers were Children’s Memorial Health Institute, Warsaw, Poland (72 pts); ASST Hospital Papa Giovanni XXIII, Bergamo, Italy (2 pts); Hannover Medical School Children’s Hospital, Germany (12 pts); Karolinska University Hospital, Stockholm, Sweden (5 pts); King’s College Hospital, London, UK (23 pts); Centro Hospitalar e Universitário de Coimbra, Coimbra, Portugal (12 pts); La Paz University Hospital, Madrid, Spain (8 pts); and OspedalePediatrico Bambino Gesu, Roma, Italy (8 pts).

Data from medical charts were analyzed, including demographic data, diagnosis, urgency of transplantation, time on the waiting list, PELD/MELD score, desensitization procedures, immunosuppression, vascular, biliary, rejection, CMV, PTLD and related post-transplant complications, and patient and graft survival. We also analyzed the results of AB0i LT according to recipient age (0–1 yr vs. older), and early and later periods of time were chosen on the basis of comparing equal numbers of patients in each period (1986–2010 vs. 2011–2018).

### 2.1. Statistical Analysis

The data were analyzed using Statistica 13.3 software, StatSoftinc (Tulsa, OK, USA). The analysis involved the assessment of baseline demographics and clinical data using median and ranges, and distributions for categorical variables. The Student *t*-test and Mann–Whitney U test were used to assess unpaired associations between continuous variables. The Chi-Square test was used to examine the differences between categorical variables. We also created Kaplan–Meier plots to analyze both patient and graft survival and compared them between groups using a log-rank test. A logistic regression analysis was used for the multivariate assessment of the relation between graft loss, recipient death, and the various predictor variables. Risk factors with a known or suspected clinical significance and a *p*-value less than 0.10 in the univariate analysis were included in the initial multivariable model. A *p*-value of less than 0.05 was considered statistically significant.

### 2.2. Ethical Approval

This study was approved by the ethics committee at the Children’s Memorial Health Institute: decision number 30/KBE/2018 issued on 10 October 2018. All participant centers followed the required ethics requisites for transmission of anonymized data to the TransplantChild, European Reference Network.

## 3. Results

Upon analysis, 142 patients with transplants between 1986 and 2018 were included in this retrospective study. There were 67 boys and 75 girls. Their ages at transplantation ranged from 0.02 to 18 yrs (median 0.93 yrs). The indications for liver transplantation were biliary atresia and other cholestatic diseases in 62 pts (43.6%), acute liver failure in 42 children (29.6%), primary hepatic tumors in 8 pts (5.6%), other diseases in 15 pts (10.6%), and re-transplantation in 15 pts (10.6%).

The transplantations were performed with grafts from 66 living and 76 deceased donors. The donor ages ranged between 0.17 and 76 yrs (median 28.2 yrs).

Follow-up after transplantation in the entire study group was 1 day to 32 yrs, with a median 4.2 yrs and a mean 5.5 ± 5.4 yrs, and was limited in most cases to transition to adult care. Overall results: patient and graft survival are shown in Figure 1 and Figure 2. Both patient and graft survival were significantly affected by deceased donor type, urgent transplantation, and the development of vascular complications (Figure 3, Figure 4 and Figure 5).

We also compared the results of AB0i LT in infants up to one year old (78 pts, Group 1) and older than one year children (64 pts, Group 2). The demographic and clinical data of recipients and donors are presented in Table 1. There were no significant differences between groups concerning pre-transplant status and urgency of transplantation, donor type and sex, pre- and post-transplant desensitization treatment, and use of induction immunosuppression. The only differences between the groups were younger donor ages for patients from group 1 and less immunosuppression given to younger children.

After transplantation, the only differences occurred in the rates of intrahepatic biliary stenosis and re-transplantation, both more common in Group 2. There was no significant difference in the rate of rejection, or patient or graft survival between the two groups, but more children survived due to re-transplantation among patients older than one year (Table 2 and Table 3, Figure 5)

There was a clear trend towards an increased number of AB0i LT over the study period. Between 1986 and 2003, seven AB0i liver transplants were performed in the eight centers participating in the study; between 1997 and 2007, there were 36 AB0i LTs; and between 2008 and 2018, the number of AB0i transplants almost tripled, reaching 99 procedures. As the oldest group had few patients and some data are lacking, we divided the patients into two almost equal groups: 72 pts transplanted between 1986 and 2010 (Group 1) and 70 pts transplanted between 2011–2018 (Group 2) to compare if there was any change in patient and graft survival over time. The group characteristics and a comparison of the results are presented in Table 4, Table 5 and Figure 6.

In the multivariable analysis, only vascular complications and year since first AB0i LT were important for graft loss and patient death (Table 6 and Table 7).

## 4. Discussion

The increasing success and utility of liver transplantations has been leading to increasingly more being undertaken internationally while organ donation is unchanged or falling, leading to an increasing discrepancy between the availability of grafts and the number of patients on waiting lists. Various responses have been developed: split liver transplantation, living donation, and wider acceptance of so-called marginal donors and grafts, including, as we described, grafts from AB0i donors. The acceptance of donor-recipient AB0 blood group incompatibility was initially limited largely to the extreme circumstances of patients with an urgent need for LT. The initial results were unsatisfactory, with high rates of rejection, HA thrombosis, biliary complications, sepsis, and other causes of graft and patient loss [5,7,8,9,10]. It has been previously noted, however, that the results of AB0i LT are better in children than in adults and are better still in the youngest children, less than one year old at transplantation. The results are also better when an AB0i graft comes from a living related donor, both in adult and pediatric recipients, particularly in the experience of Japanese centers [2,5,11,12]. However, there are no studies considering large numbers of patients from several major centers in Europe.

Therefore, we aimed to consider the achievements of centers in Europe, to review current practices, and to set a benchmark for the use of this source of grafts for liver transplantation in children.

### 4.1. Immunosuppression and Rejection

During our study period, several protocols of pre-transplant desensitization were developed internationally in an attempt to prevent early or immediate humoral rejection and to improve late results after AB0i LT: splenectomy, plasma exchange, immunoadsorption, high-dose intravenous immunoglobulins (IVIG), and antiCD20 monoclonal antibodies (rituximab) [7,8,9,11,13,14,15,16,17,18]. We found that the pre-transplant desensitization of children before AB0i LT was very uncommon in European centers, with high-dose IVIG being given more often than alternatives but only in 8% of recipients. We speculate that the low age of the children included in the study and that the production of anti-AB isoagglutinins is age dependent, resulting in low titers of anti-AB antibodies in young children, very often lower than the goal of desensitization (titer 1:16 and lower). This phenomenon is probably responsible for the extremely low incidence of hyperacute or early AMR in infants and children younger than 2 years undergoing AB0i LT [5,6,11,12,13,19].

However, almost 30% of children were treated with IVIG postoperatively. Other methods of desensitization were used only occasionally both pre- and post-transplantation. As reported in the literature, pediatric immunosuppression protocols for AB0i liver transplant differ significantly between centers [4,6,10,11,17,19,20,21,22,23,24,25,26]. There was no single protocol of post-transplant immunosuppression used by centers in our series; however, about 70% of patients received induction with antiIL-2 CD25Ab (basiliximab and daclizumab). Most guidelines published in the literature support the inclusion of steroids in the IS protocol after AB0i LT at least for a few months after transplantation [2,11,12,19,23].

When desensitization and immunosuppression between the two age groups were compared, children below 1 year of age received less immunosuppressive drugs after AB0i LT (Table 1), which was the only significant difference. Surprisingly, we did not find any difference in the pre- and post-AB0i LT immunosuppressive treatments when we compared two study periods (1986–2010 vs. 2011–2018), with the exception of an increased use of antiCD25 induction therapy in the later period.

Isoagglutinin levels increase with age from infancy so we expect less acute rejection episodes and severity in children below 1 year of age, particularly of the humoral type [11,12,27,28]. Most of the acute rejection episodes that were confirmed by biopsy in our series occurred within 12 months following transplantation. There were, however, no differences between infants and older children in the incidence of acute rejection including antibody mediated rejections at any time point after transplantation. Three infants (Group 1) and four children aged >1 yr (Group 2), developed antibody-mediated rejection (AMR) treated with steroid boluses and with IVIG, antilymphocytic globulin, rituximab, or plasmaphereses. Chronic rejection was found in 4 pts from Group 1 and in only 1 pt from Group 2. None of these differences were statistically significant.

### 4.2. Complications

It is recognized that patients receiving AB0i grafts are at increased risk of developing certain complications [7,10,13,29,30,31]. Hyperacute or early humoral rejection resulting in massive intrahepatic microcirculation endothelial damage and thrombosis may cause hemorrhagic graft necrosis and loss within a few days. It is fortunately not common in pediatric patients but may also contribute to the increased incidence of hepatic artery thrombosis, which may develop in as many as 24% of AB0i graft recipients [31,32]. In our series, only 10% of recipients developed HAT. In another 13.3% patients, PVT occurred. More children less than 1 year old presented vascular complications, which did not reach statistical significance. However, thrombosis may be the result of technical difficulties rather than the consequence of immunological AB0 mismatch, as children below 1 year would not be expected to suffer increased risk of humoral rejection after AB0i LT.

Another common complication that may occur usually within the first 3 months after AB0i LT is damage to the intrahepatic bile ducts and development of multiple intrahepatic biliary stenosis. It may be caused by immunological insult by humoral reaction to the donor blood group antigens, which are present on the epithelium of bile ducts for 3–6 months after LT [28,32,33]. In this study, we observed this cholangiopathic complication significantly more often in children over 1 year of age, which further supports the observation of the safe use of AB0i LT in infants. This complication was recognized only in patients transplanted before the year 2011, associated with a more frequent acute rejection rate. There was no differences between age groups or in relation to the period of transplantation in the rate of biliary anastomotic stenosis.

### 4.3. Patient and Graft Survival

The overall patient and graft survival in our series is perhaps not as good as we wish, but our study period included children with transplants from as early as 1986. Comparing these results to those reported by the European Liver Transplant Registry (ELTR), for a similar period, the results in our series of AB0i LT are similar to all contemporaneous pediatric liver transplantations reported to the ELTR (Figure 7) [34].

Although both patient and graft survival are better in younger children [5,6], the difference is not statistically significant in our series. Similar results, however, came about with significantly increased re-transplantation rates in the older group. More patients being transplanted due to acute liver failure partially explains the increased mortality in the older group, while the more common development of cholangiopathic complications could contribute to increased graft loss in this group. This latter complication was not observed after 2011 in any patient, and it was connected with significant improvement in overall patient and graft survival in the second study period, when significantly less re-transplantations were also necessary. A general improvement in pediatric liver transplantation was observed over this time [34], but the reduction in re-transplantation resulted mainly from the better survival outcomes of patients transplanted electively. Meanwhile, among urgent transplantations, the improvement is small, suggesting that, for urgent LTs, factors other than AB0 incompatibility could play an important role in mortality.

We used a univariate analysis as a starting point for the creation of a multivariate logistic regression model to assess the relationship between graft loss, recipient death, and the various predictor variables. We showed that graft loss and recipient death were only related to two independent risk factors: vascular complications and time from first transplant, supporting the view that immunological reactions do not have a major influence on the results of AB0i LT in our particular pediatric populations.

In summary, our study has shown that the outcome of AB0i liver transplantations in children has improved considerably over the last 20 years. The reasons for this improvement are probably multifactorial, including better perioperative and anesthetic management; better perioperative monitoring; increasing multidisciplinary experience and surgical skills; improved immunosuppression and understanding; and the prevention, diagnosis, and treatment of humoral rejection (CD4, DSA) [6,7,11,19,35,36,37,38]. The role of desensitization before AB0i LT in pediatric recipients is still not defined. Although our study showed that it is not widely used in children and is more often preferred in combination therapy with proven AMR after transplantation in the literature [15,16,25,26,27,32,39], the consensus for post-transplant immunosuppression in our data suggests a tendency towards sequential immunosuppression with induction by antiCD25 antibodies and triple drug therapy with tacrolimus, mycophenolate mofetil, and steroids for at least a few months after LT, followed by a slow reduction in IS thereafter [2,6,12,16,24,25]. However, good results were also reported with standard IS, as used in AB0 compatible transplantations [29]. Future strategies including the prevention of complement mediated cell damage (eculizumab) or AMR (bortezumib) may be introduced to improve the results of AB0i LT further, particularly among older recipients [25].

## 5. Conclusions

While there is a pressing need for expansion of the pediatric donor pool in response to a critical organ shortage, the improvement in the results of AB0i LT makes it justifiable to use this type of transplantation not only in circumstances of emergency transplantation due to acute liver failure and severe acute decompensation of chronic liver disease and semi-elective transplantation for primary hepatic tumors but also in elective transplantations, particularly from living related donors [1,2,3,25,39,40]. The role of pre-transplant preparation and post-transplant immunosuppression remains to be established optimally in multicenter prospective studies.

## Figures and Tables

**Figure 1 children-08-00760-f001:**
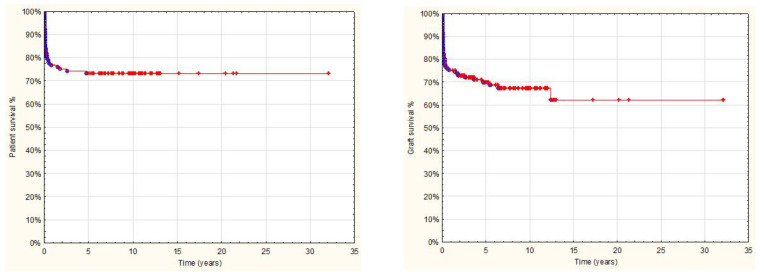
Overall AB0i LT patient and graft survival (142 pts after AB0i LT; 1986–2018).

**Figure 2 children-08-00760-f002:**
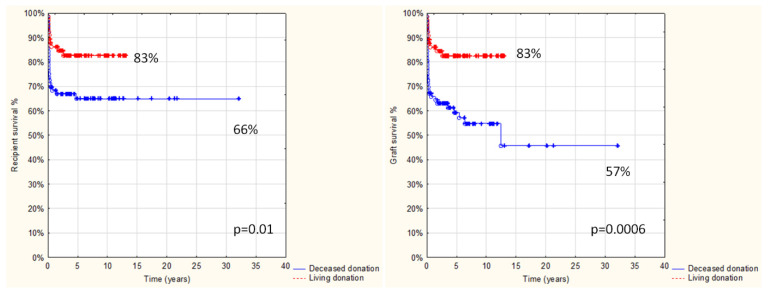
Patient and graft survival in relation to donor type in 142 pts after AB0i LT (*p* < 0.05).

**Figure 3 children-08-00760-f003:**
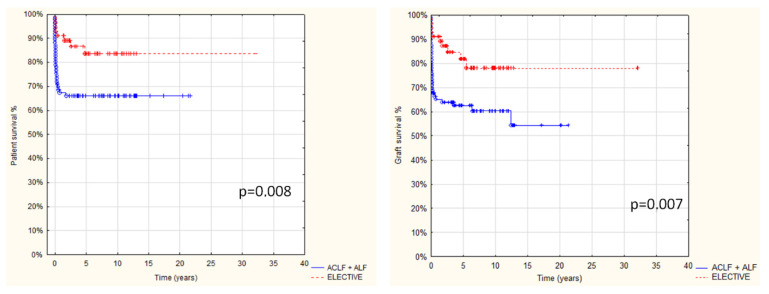
Comparison of patient and graft survival in relation to transplant urgency: elective vs. urgent (including acute-on-chronic and acute liver failure) in 142 pts after AB0i LT (*p* < 0.05).

**Figure 4 children-08-00760-f004:**
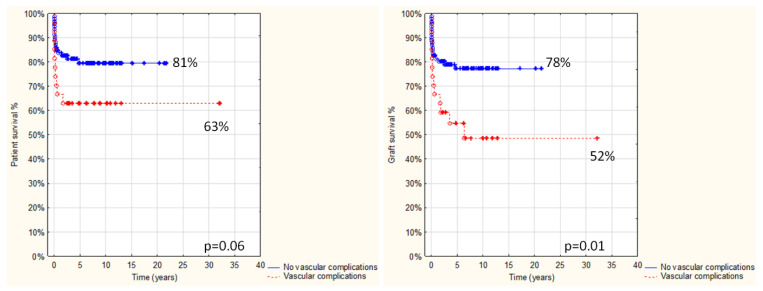
Vascular complications, and patient and graft survival in 142 pts after AB0i LT (*p* < 0.05).

**Figure 5 children-08-00760-f005:**
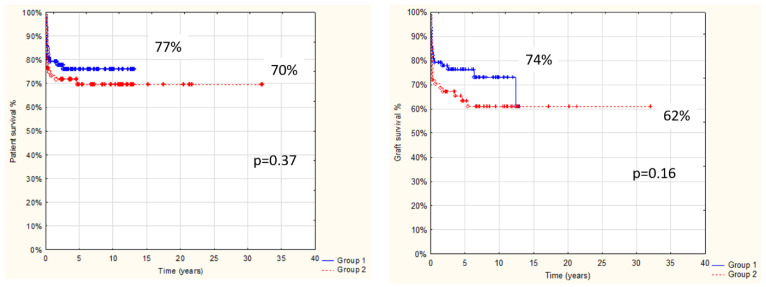
Patient and graft survival; comparison of children after AB0i LT aged ≤ 1 yr (Group 1) and older (Group 2).

**Figure 6 children-08-00760-f006:**
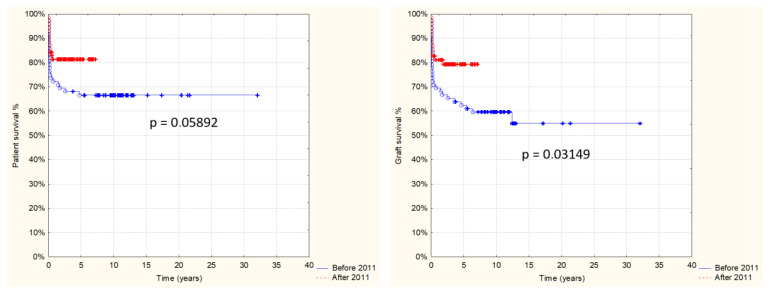
Patient and graft survival after AB0i LT, comparison between the periods 1986–2010 (Group 1) and 2011–2018 (Group 2).

**Figure 7 children-08-00760-f007:**
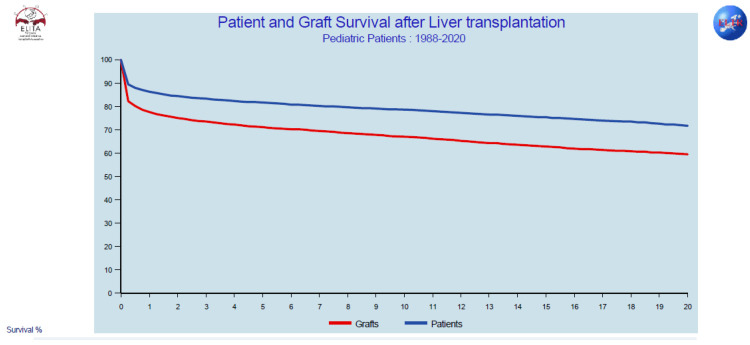
Overall patient and graft survival for pediatric LT included in the ELTR database between 1988 and 2020.

**Table 1 children-08-00760-t001:** Demographic and clinical data of recipients and donors. Data divided for pts ≤ 1 yr of age (Group 1) and older than one yr (Group 2). Data corrected for availability.

Characteristics		Group 1 (*n* = 78)		Group 2 (*n* = 64)		*p*-Value
		Tx ≤ 1 yr		Tx > 1 yr		
		N	%	N	%	
**Age (years): Range; Median **		**0–1; 0.6 **		**1–18; 3.8**		
Recipient sex	male	35	45%	32	50%	n.s.
	female	43	55%	32	50%	
Urgency of LT	elective	30	38%	26	40%	n.s.
	urgent	24	31%	30	47%	*p* < 0.05
	decompensation	24	31%	8	13%	*p* < 0.01
Days on waiting list: range; median		1–272; 25		0–831; 12		n.s.
PELD	<9	7	11%	15	33% (*p* < 0.02)	n.s. (for whole groups median values)
	10–19	14	21.5%	8	17% (n.s.)	
	20–29	17	26%	10	22% (n.s.)	
	30–39	14	21.5%	7	15% (n.s.)	
	>40	13	20%	6	13% (n.s.)	
	NA	13	17%	18	28% (n.s.)	
Donor age: range; median		0–68; 27		0–76; 31		*p* = 0.002
Living donor		41	53%	25	39%	n.s.
Deceased donor		37	47%	39	61%	
Donor sex	male	30	43%	34	54%	n.s.
	female	40	57%	29	46%	
	NA	8	10%	1	1.5%	
Pre-transplant treatment	pre-LT plasmapheresis	3	4%	4	6%	n.s.
	pre-LT immunoglobulins iv	2	3%	5	8%	n.s.
	pre-LT immunoadsorption	0	0%	1	2%	n.s.
	pre-LT rituximab	3	4%	1	2%	n.s.
Induction IS	induction (basiliximab or daclizumab)	52	67%	46	72%	n.s.
Post-transplant treatment	post-LT plasmapheresis	2	3%	4	6%	n.s.
	post-LT immunoglobulins iv	19/68	28%	18/61	29%	n.s.
	post-LT immunoadsorption	0	0%	4	6%	
Total IS	≤2 drugs	8/63	13%	1/49	2%	*p* = 0.04
	>2 drugs	55/63	87%	48/49	98%	
	NA	15	19%	15	23%	

Abbreviations Table 1: Tx—transplantation, yr—year, NA—status unknown, pre-LT—before liver transplantation, post-LT—after liver transplantation, iv—intravenous, n.s.—not significant.

**Table 2 children-08-00760-t002:** Results of AB0i LT—comparison of recipients ≤ 1 yr of age and older. Data corrected for availability (removal of missing data).

Characteristics	Group 1 (*n* = 78)		Group 2 (*n* = 64)		*p*-Value
	Tx ≤ 1 yr		Tx > 1 yr		
	N	%	N	%	
Biliary complications	9/63	14%	11/51	22%	n.s.
*biliary anastomotic stenosis*	9	14%	9	18%	n.s.
*biliary intrahepatic stenosis*	0	0%	5	10%	*p* = 0.01
*vanishing bile ducts*	0	0%	1	2%	n.s.
*biliary NA*	15	19%	13	20%	
Vascular complications	18/65	28%	9/55	16%	n.s.
*HAT*	8	12%	4	7%	n.s.
*PVT*	11	17%	5	9%	n.s.
*Vascular NA*	13	17%	9	14%	
PTLD	3	4%	4	6%	n.s.
CMVdisease	21/64	33%	11/51	21%	n.s.
*CMV NA*	14/78	18%	13/64	20%	
Graft loss	20	26%	24	37%	n.s.
ReTx	4	5%	12	19%	*p* = 0.01
Death	18	23%	19	30%	n.s.
Post-LT follow up (months): range; median	0–157; 40		0–385; 59		n.s.

Abbreviations Table 2: Tx—transplantation, yr—year, NA—status unknown, pre-LT—before liver transplantation, post-LT—after liver transplantation, HAT—hepatic artery thrombosis, PVT—portal vein thrombosis, CMV—cytomegalovirus, n.s.—not significant, PTLD—post-transplant lymphoproliferative disease, ReTx—re-transplantation.

**Table 3 children-08-00760-t003:** Rejection rate. Results of AB0i LT—comparison of recipients ≤ 1 yr of age and older (corrected for mortality).

Characteristics	Group 1 (*n* = 78)		Group 2 (*n* = 64)		*p*-Value
	<1 yr		>1 yr		
	N	%	N	%	
AR < 3 months	12	16%	13	21%	n.s.
AR > 3 months	14	22%	8	16%	n.s.
Chronic rejection	4	5%	1	1.6%	n.s.

Abbreviations Table 3: yr—year, AR—acute rejection, n.s.—not significant.

**Table 4 children-08-00760-t004:** Characteristics of 72 pts transplanted before 2011 (Group 1) and 70 pts transplanted since 2011 (Group 2). Data corrected for availability (removal of missing data).

Characteristics		Group 1 (*n* = 72)		Group 2 (*n* = 70)		*p*-Value
		(1986–2010)		2011–2018		
		N	%	N	%	
Age years: range; median		0–18; 1.3		0–17.6; 0.9		*p* < 0.025
Recipient sex	Male	31	43%	36	51%	n.s.
	Female	41	57%	34	49%	
Urgency of LT	Elective	24	33%	32	46%	n.s.
	Urgent	38	53%	16	23%	*p* = 0.0002
	Decompensation	10	14%	22	31%	*p* < 0.02
Days on waiting list: mean; range; median		62; 0–821; (9)		57; 0–307; (37)		*p* < 0.03
PELD CATEGORY (points)	BELOW 9	11	20%	11	20%	n.s.
	10 to 9	12	22%	10	18%	
	20 to 29	14	25%	13	23%	
	30 to 39	8	15%	13	23%	
	≥40	10	18%	9	16%	
	NA	17	31%	14	25%	
Donor age (yrs): range; median		0–76; 26.8		0–76; 30		n.s.
Living donor		29	40%	37	53%	n.s.
Deceased donor		43	60%	33	47%	
Donor sex	Male	34	48%	30	48%	n.s.
	Female	37	52%	32	52%	
	NA	1	1.4%	8	13%	
Induction treatment	basiliximab/daclizumab	41	73%	57	84%	n.s.
Pre-transplant treatment	plasmapheresis	2	3%	5	7%	n.s.
	IVIG	4	6%	3	4%	n.s.
	immunoadsorption	0	0%	1	1%	n.s.
	rituximab	1	1%	4	6%	n.s.
Post-transplant treatment	plasmapheresis	1	1.5%	3	5%	n.s.
	IVIG	19	29%	18	29%	n.s.
	immunoadsorption	3	4%	1	2%	n.s.
Total IS	≤2 drugs	4	7%	5	9%	n.s.
	>2 drugs	51	93%	52	91%	
	NA	17	31%	13	23%	

Abbreviations Table 4: N—number, LT—liver transplantation, NA—status unknown, IVIG—intravenous immunoglobulins, IS—immunosuppression, n.s.—not significant, PELD—pediatric end stage liver disease score.

**Table 5 children-08-00760-t005:** Post-LT follow-up of 72 pts transplanted before 2011 (Group 1) and of 70 pts transplanted since 2011 (Group 2). Data corrected for availability (removal of missing data).

Characteristics	Group 1 (*n* = 72)		Group 2 (*n* = 70)		*p*-Value
	1986–2010		2011–2018		
	N	%	N	%	
Biliary complications	10	17%	18	18%	n.s.
biliary anastomotic stenosis	8	15%	10	18%	n.s.
biliary intrahepatic stenosis	5	9%	0	0%	*p* < 0.04
vanishing bile ducts	1	2%	0	0%	n.s.
biliary NA	13	22%	15	27%	n.s.
Vascular complications	14	23%	13	22%	n.s.
HAT	6	10%	6	10%	n.s.
PVT	9	15%	7	12%	n.s.
vascular NA	10	16%	12	21%	n.s.
AR	29/71	41%	16/69	23%	*p* < 0.03
PTLD	3	5%	4	7%	n.s.
CMV disease	13	22%	19	33%	n.s.
ReTx	12	17%	1	1.5%	*p* = 0.002
Death	24	33%	13	19%	*p* < 0.05
Post-Tx follow up (months): range; median	0–385; 105		0–86; 35		*p* < 0.0001

Abbreviations Table 5: N—number, NA—status unknown, HAT—hepatic artery thrombosis, PVT—portal vein thrombosis CMV—cytomegalovirus, n.s.—not significant, PTLD—post-transplant lymphoproliferative disease, ReTx—re-transplantation.

**Table 6 children-08-00760-t006:** Risk factors for graft loss. Multivariable analysis (logistic regression): only factors with *p*-value < 0.1 according to a univariate analysis were included.

Variables	Adjusted OR (95% CI)	*p*
Acute—urgent LT	1.35 (0.34–5.3)	0.663
Elective LT	0.65 (0.18–2.32)	0.514
Deceased donor	2.14 (0.64–7.14)	0.215
Vascular complications	4.74 (1.68–13.33)	0.003
Waiting time on list (days)	0.99 (0.994–1.005)	0.782
Years from 1st ABOi	0.926 (0.853–1.005)	0.064

Abbreviations: LT—liver transplantation.

**Table 7 children-08-00760-t007:** Risk factors for recipient death. Multivariable analysis (logistic regression): only factors with *p*-value < 0.1 according to a univariate analysis were included.

Variables	Adjusted OR (95% CI)	*p*
Acute—urgent LT	1.24 (0.32–4.82)	0.759
Elective LT	0.55 (0.15–1.93)	0.348
Deceased donor	1.31 (0.40–4.29)	0.651
Vascular complications	2.80 (1.03–7.61)	0.043
Waiting time on list (days)	0.99 (0.994–1.005)	0.799
Years from 1st AB0i	0.944 (0.874–1.019)	0.142

Abbreviations: LT—liver transplantation.

## Data Availability

Most relevant data are within the paper. Most of the data were taken from patients’ medical records. Since the data collected in the medical records are sensitive and, thus, protected by the law (the Act on Personal Data Protection and the Medical Records Act), access to these data is limited to healthcare professionals employed in CMHI and other centers that took part in this study.

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
