# Peer review of "ABO Incompatible Liver Transplantation in Children: A 20 Year Experience from Centres in the TransplantChild European Reference Network"

_children, 2021, doi:10.3390/children8090760_

Round 1

Reviewer 1 Report

Dear authors thank you for letting me participate in the review of your manuscript "AB0 incompatible liver transplantation in pediatric population – European experience. Report of TransplantChild, The European Reference Network on Pediatric Transplantation" submitted to Children. 

While you report very interesting Data to improve Pediatric Liver Transplantation in the Situation of increasing Organ shortage I would suggest to work on your methods part and hence the Quality of your data report.

I point out some of the main concerns with regards to your manuscript. Especially why was the time period Chosen arbitrarily to have the same amount of n, wouldn't it be more interesting to consider other factors surgical consistancy multidisciplinary approach? Also it would be important to comment on the rejections and n of protocol biopsies as well as overall biopsies taken to better understand your results. 

Author Response

Thank you very much for reading the manuscript and valuable comments.

Concerning arbitrary chosen time period our aim was to compare numerically similar groups. On the other hand the group I had only 1 pt transplanted in 1986 and then great majority of patients were transplanted between 1995 and 2010. W added to the Material and Methods a sentence explaining this choice:

Lines 90-92:  ..”We also analyzed results of AB0i according to recipient age (0-1yr vs. older); and in two periods of time chosen on the basis of creating equal number of patients in both groups (1986-2010 vs. 2011-2018).”

And more explanation is presented in Results:

Lines 166-173

There was a clear trend towards increased number of AB0iLT over the study period. Between 1986-2003, 7 AB0i liver transplants were performed in the 8 centers participating in the study, between 1997-2007 there were 36 AB0i LT and between 2008-2018 number of AB0i transplants almost tripled reaching 99 procedures. As the oldest group is small and some data are lacking, we divided patients in to two almost equal groups: 72 pts transplanted between 1986-2010 (Group 1) and 70 pts transplanted between 2011-2018 (Group 2) to compare if there was any change in patient and graft survival over time. Group characteristics and comparison of results are presented in tables 4, 5 and figure 6.

Although the study periods were created arbitrary in uniovariate analysis we found that the results depend on the years from first AB0iLT independently of particular borderline year.

Concerning rejection rate there is comparison on AR incidence between recipients younger than 1 year and older (Lines 149-143 and Table 3)

After transplantation the only differences occurred in incidence of intrahepatic biliary stenosis and retransplantation rate, both more common in group 2. There was no significant difference in the rate of rejection, patient or graft survival between the two groups, but more children survived due to retransplantation among patients older than  one year (table 2, 3, figure 5)

We added also information on incidence of AR in groups compared due to transplantation period (Table 5).

We analysed in univariate and multivariate analysis factors influencing graft and patient survival and AR was not found to be signifficant, but in Disscussion we added sentence:

…”This complication was recognized however only in patients transplanted before year 2011, together with more frequent acute rejection rate in this study period. (Lines 269-271)

We have no data on protocol biopsies in these patients, most centers did not perform it, so we could not analyse results of these biopsies.

We also developed conclusions with adding a sentence

…“The role of pretransplant preparation and posttransplant immunosuppression should still be established probably in multicenter prospective studies.  (Lines 339-341)

Reviewer 2 Report

The authors present an important experience from 8 European pediatric liver transplantation centers on ABO-incompatible liver transplantation. The study analyzed the patient and graft survival based on donor type, emergency, vascular complications, results of this type of LT based on the age (under one year or over) or the historical period (before and after 2010). The results of this study are important in supporting the idea that this kind of LT should be used more frequently and not only in emergencies as acute liver failure, acute on chronic, or as semi-elective transplantation due to primary hepatic tumor.

The paper is well written, clearly presenting the background, the aim, the methods, results and discussed the findings appropriately in the light of other publications in the field.

Just some minor changes should be made:

  • if abbreviations were defined, it is better to use them for all situations in the text;
  • there is no p-value or ns for PELD analysis in table 1;
  • some minor editing - line 251 - "aged less than 1 year of age" should be "less than one year old" maybe.

Author Response

Thank you very much for reading the manuscript and valuable comments. According to the comments we corrected editig and language errors and standardized the abbreviations; we checked differences in PELD values between the study groups and included it to the table 1.